# Experimental Investigation of Mechanical Property and Wear Behaviour of T6 Treated A356 Alloy with Minor Addition of Copper and Zinc

**Nithesh Kashimat, Sathyashankara Sharma, Rajesh Nayak, Karthik Birur Manjunathaiah, Manjunath Shettar *** and Gowrishankar Mandya Chennegowda ***

Department of Mechanical and Industrial Engineering, Manipal Institute of Technology,
Manipal Academy of Higher Education, Manipal 576104, India
* Correspondence: manjunath.shettar@manipal.edu (M.S.); gowri.shankarmc@manipal.edu (G.M.C.)

**Abstract:** The present study examines the effect of trace additions of copper (up to 1 wt.%) and zinc (0.5 wt.%) as the alloying elements on the microstructure, hardness, and wear behaviour of T6 treated A356 (Al-7Si) alloy. Wear tests were conducted using a pin-on-disc tribometer under a constant sliding speed of 200 RPM, varying applied load (20–40 N), and sliding distance (0–3000 m) to determine the wear rate and the coefficient of friction. The results indicated a minimum of 1 wt.% of copper was required to form the $Al_2Cu$ intermetallic phase, resulting in a finer grain structure and improved hardness. However, the role of zinc as a trace element was not observed on the microstructure; the observed changes may be the combined effect of copper and zinc as a whole. The highest hardness of 107 VHN (98% increase) was achieved with 1 wt.% copper addition during peak aging at 100 °C. Also, wear tests showed that adding 1 wt.% copper to the A356 alloy and a 100 °C precipitation hardening (T6) treatment improved the wear resistance by 150–182% with a reduced coefficient of friction.

**Keywords:** Al-7Si; T6 treatment; microstructure; hardness; wear

## 1. Introduction

Aluminium alloys are considered one of the most economical, versatile, and attractive materials, with a wide range of applications from basic ductile foils to high-end engineering applications, making them the second most used materials next to steel in structural metals. Over the years, researchers have been inspired to study the development of lightweight materials that can be used as an alternative in manufacturing various automobile and aerospace components. It is known that lightweight components made of cast aluminium alloy have been used extensively in the automobile and aerospace sector, mainly due to their outstanding physical, mechanical, and tribological properties, providing cost effective alternatives to the conventional use of cast iron in engine blocks and pistons [1–3]. Appropriate alloying and heat treatment can lead to enhanced properties in aluminium and its resistivity to progressive oxidation to form rust. An increase in the volume fractions of the intermetallic phases formed by the alloying element and the elemental silicon constituent formed during solidification or age hardening significantly improves the Al-Si tribological properties of alloys. Researchers have found that silicon addition improves mechanical and wear properties up to its eutectic composition, above which it has a negative impact on these properties. As a result, Al-Si hypoeutectic alloy (A356) finds a key place in today's industrial world for usage in aircraft and automotive applications [4–6]. Typical applications of A356 alloy include aircraft engine and pump parts, airframes and landing wheels, truck chassis parts, aircraft fittings, and control parts, along with structural elements which require high strengths [4,7–10]. Regardless of its excellent properties, there is also a provision to improve its hardness along with its wear properties. The performance of these alloys in terms of wear resistance is influenced by factors such as

the microstructure, which includes aspects such as the shape, size, composition, and distribution of its micro constituents; their mechanical properties, including hardness, toughness, maximum tensile strength, and flexibility; and the conditions in which they are used, such as the type of load, the sliding speed, temperature, environmental factors, and the material of the opposing surface [11–14]. The addition of various alloying elements such as magnesium (Mg), copper (Cu), and zinc (Zn) has two to three-fold advantages in the mentioned properties, and also influences wear behaviour in A356 alloy due to precipitation hardening and solid solution strengthening [15]. In addition, these alloying elements such as Mg, Cu, and Zn, present in small quantities, enter into Al solid solution forming various strengthening phases such as $Al_5Cu_2Mg_8Si_6$, $Al_5FeSi$, $Al_8Mg_3FeSi_6$, $Al_{15}(Mn, Fe)_3Si_2$, and/or other particles under different conditions [16–18]. Several additional phases, including $CuMgAl_2$, $Mg_2Si$, and $CuAl_2$ exist in metastable conditions in Al-Si-Mg-Cu alloys [19]. During heat treatment, Zn promotes the precipitation of intermetallics of the $Mg_3Zn_3Al_2$ and $Mg_2Zn$ types. Researchers have conducted several studies in optimizing and improving tribological properties in Al-Si alloys. T6 artificial aging treatment is one of the thermal treatments most frequently used to increase the strength of A356 cast alloys. The T6 heat treatment aims to provide an optimum combination between strength and ductility in the A356 alloy. With aging, the alloy not only becomes stronger and more ductile, but also more prone to stress corrosion. The strength-enhancing effect of aging results from the decomposition of a semi-stable, over-saturated solid solution created through solutionising and rapid quenching. Previous research has shown that the strength of Al-Si alloys can be influenced by a variety of factors such as grain refinement, chemical composition, hardening conditions, the size and shape of eutectic Si particles in the structure, and its dendritic arm spacing [20–24].

Few of the studies on the wear-related properties of aluminium alloys were conducted by previous researchers. Colak et al., studied the wear and frictional properties of pure aluminium alloy where Cu was added to commercially available aluminium in 2, 4, 6 and 8 wt.%. Results found that the specific wear rate decreased from $10^{-12}$ to $10^{-13}$ m²/N range with the increase in Cu wt.%. Also, at low load conditions, the coefficient of friction was reduced by 84% with the increase in Cu wt.% [25]. Hassan et al. conducted wear tests on Al-Mg-Cu alloy and discovered that the addition of SiC particles significantly improved the wear properties of the alloy. With increasing sliding distance, the wear volume loss of alloys increased linearly. Additionally, they found that the wear resistance of the Al-Mg-Cu alloy increased with up to 5 wt.% Cu addition, but significantly affected the coefficient of friction values [26]. Tuti Y. Alias and M.M. Haque examined the wear properties of Al-Si eutectic alloy in its as-cast and heat-treated forms using a pin-on-disc tribometer. They found that both as-cast and heat-treated alloys showed an increase in wear rate with an increase in input weight, rotation speed, and sliding distance. However, the as-cast specimen showed greater wear than the heat-treated sample due to the inherent characteristics of the alloy [27]. Sudarshan and Surappa studied the sliding wear behaviour of A356-(6–12 vol.%) fly ash reinforced composite using a varying load of 10–80 N at a constant sliding speed of 1 m/s. The wear rate graph obtained from pin-on-disc apparatus was verified using the weight loss method. Authors found that at low loads up to 20 N, the wear rate of unreinforced alloy and composites was approximately similar. On the other hand, at higher load conditions, wear rate increased with increasing loads mainly due to curling effects in graphical method. Results also showed that the wear rate decreases with increasing vol.% of the reinforcement added [28]. Prabhudev et al. found that 0.5 wt.% copper addition to A356 improved the wear resistance property of the alloy. Also, copper addition promoted the formation of an oxide layer between the mating surfaces resulting in improved wear resistance and friction. Under all loading circumstances, the frictional force increased as the normal pressure and sliding distance increased [29]. In addition, Kori and Prabhudev found that Cu addition to A356 alloy resulted in the formation of an iron-oxide layer between mating surfaces which decreased the wear rate of the alloy. In A356 alloy, abrasive/oxidative and delamination wear mechanisms largely dominated,

whereas in 0.5 wt.% Cu added A356 alloy, abrasive/oxidative and Fe-rich oxide layer was dominant [30]. Further, according to a study conducted by Kori et al., the addition of a minimal amount of magnesium (ranging from 0.3 to 1 wt.%) to the A356 alloy caused a change in the acicular eutectic silicon structure, transforming it into a lamellar and fibrous structure, while the α-Al remained unchanged [15].

However, limited research has been carried out concerning the minor addition of copper, zinc, and magnesium to A356 alloy for the improvement of its wear property. Therefore, in the present work, an attempt is made by incorporating a small amount of copper, zinc, and magnesium as an alloying element into A356 to determine the change in microstructure, hardness, and wear behaviour of both as-cast and age hardened conditions.

## 2. Methodology

### 2.1. Preparation of Alloys

A356 alloy ingots for the current work were procured from Laxmi Metal Exchange, Coimbatore, India and a chemical analysis was performed in accordance with ASTM E-1251-2011 standards. The procured zinc and copper powders were subjected to particle size analysis. An amount of 90% of the total volume in zinc powder was between 20 and 100 μm with an average particle size of 33 μm, and 90% of the total volume in copper powder was between 5 and 30 μm with an average particle size of 9.33 μm. Table 1 lists the prepared alloys and their designations used in this work. The chemical composition of the prepared as-cast alloy with small weight percentages of magnesium, zinc, and copper is shown in Table 2. All the alloys were fabricated using a two-step stir casting method. A small amount of alkaline powder and solid hexacholoroethane (degassing agent) was added during fabrication to eliminate the formation of slag and oxides. All the as-cast bar specimens were cut into smaller samples with different dimensions (10–14 mm) using wire EDM machining process, as shown in Figure 1, and used for characterization.

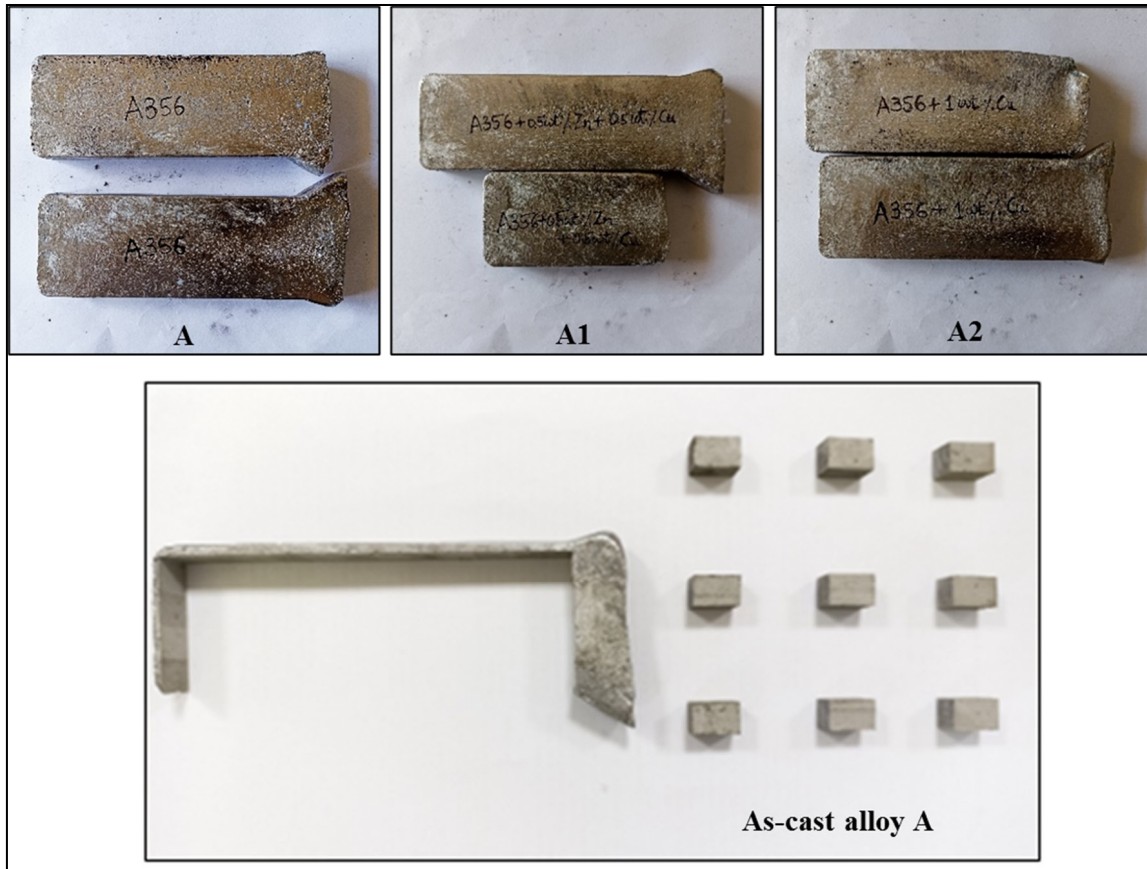

**Figure 1.** As-cast bars and wire EDM cut samples of alloy A.

**Table 1.** Casted alloys and their designations.

| Casting | Designation |
|---|---|
| A356 + 1 wt.% Mg | A |
| A356 + 1 wt.% Mg + 0.5 wt.%Zn + 0.5 wt.% Cu | A1 |
| A356 + 1 wt.% Mg + 1 wt.% Cu | A2 |

**Table 2.** Alloys and their chemical composition (wt.%).

| Designation | Si | Mg | Cu | Zn | Fe | Others | Al |
|---|---|---|---|---|---|---|---|
| A356 | 6.5 | 0.2 | 0.2 | 0.1 | 0.2 | 0.1 | Bal. |
| A | 6.5 | 1.2 | 0.2 | 0.1 | 0.2 | 0.1 | Bal. |
| A1 | 6.5 | 1.2 | 0.7 | 0.6 | 0.2 | 0.1 | Bal. |
| A2 | 6.5 | 1.2 | 1.2 | 0.1 | 0.2 | 0.1 | Bal. |

### 2.2. Age Hardening Treatment of A, A1, and A2 Alloys

The prepared as-cast A, A1, and A2 specimens were initially homogenized at 520 °C for 8 h, followed by hot water quenching at 60 °C, and artificial aging at 100 and 200 °C separately for different time intervals to determine the peak hardness value. The surfaces were polished before hardness testing to remove any oxide layers or surface impurities that might have occurred during the heat treatment procedure.

### 2.3. Characterization of A, A1, and A2 Alloys

The mirror-finished samples were subjected to microstructure examination using a scanning electron microscope (SEM). The intermetallic phases formed were identified using X-ray diffraction (XRD) analysis [31].

Hardness measurements were carried out as per ASTM E384 standard using Matsuzawa micro Vickers hardness tester at room temperature with a load of 200 gmf and dwell time of 15 sec. Vickers hardness number (VHN) was calculated by averaging 5 concurrent hardness numbers obtained from the indentations.

### 2.4. Wear Test of A, A1, and A2 Alloys

From the prepared castings A, A1, and A2, wear specimens of different diameters (6, 8, and 10 mm) and heights of 30 mm were machined using the wire EDM process. The obtained A, A1, and A2 pin specimens were subjected to a dry sliding wear test according to ASTM G99 standard with varying applied load and sliding distance at room temperature. Figure 2 shows the as-cast wear specimens used to conduct wear tests.

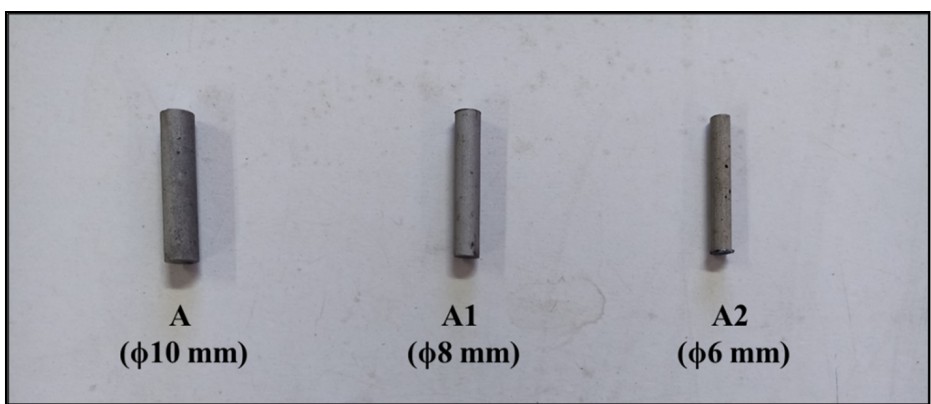

**Figure 2.** As-cast wear specimens.

Wear specimens were placed in a collet, which was fixed to the holder during wear tests. The current experiment used an EN-32 steel disc of 165 mm in diameter and 8 mm thickness, with a roughness value of 0.47–0.87 m and a hardness of HRC 61. Both the steel disc and the wear specimens were treated with emery paper and acetone prior to each test to preserve dry sliding conditions. Further, wear tests were carried out on the pin-on-disc tribometer with a track diameter range of 0–90 mm and a maximum loading capability of 200 N. Figure 3 shows the pin-on-disc tribometer used to conduct the wear test. Technical specifications of the tribometer used can be seen in Table 3.

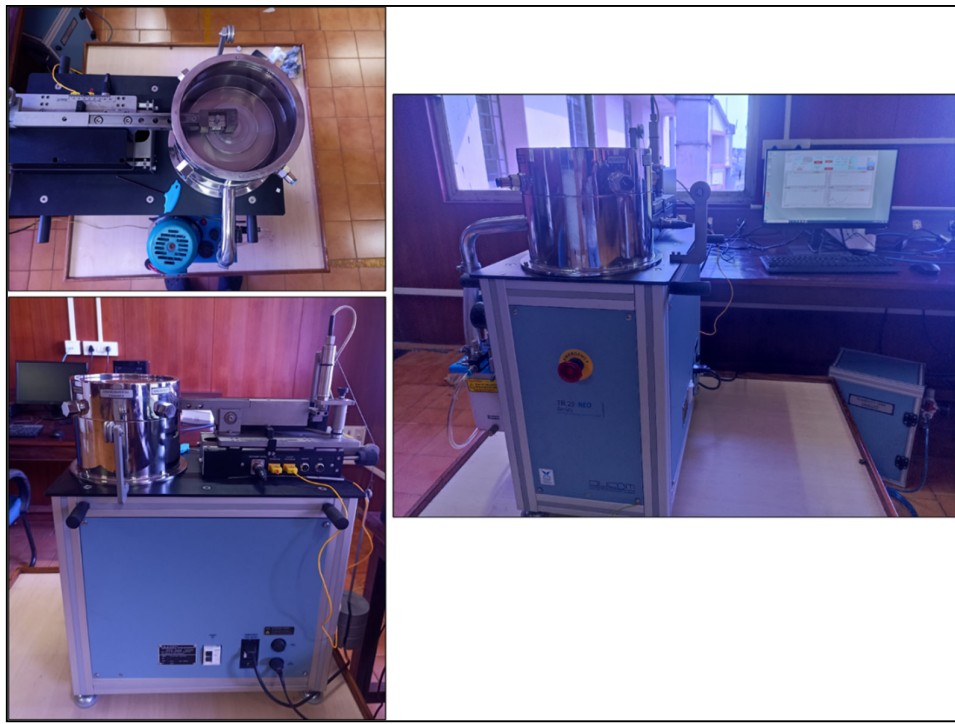

**Figure 3.** Pin-on-disc tribometer.

**Table 3.** Specifications of tribometer.

| Model | TR 20 Neo |
|---|---|
| Disc size | Φ100 × 6 mm thickness |
| Material | Hardened ground steel (EN-31), 65 HRC |
| Rotational Speed | 80 rpm (min.) and 800 rpm (max.) |
| Motor | AC motor, 230 V, 0.37 kW |
| Wear track diameter | 20 mm (min.) and 80 mm (max.) |
| Normal load | Upto 200 N max |

The wear behaviour of as-cast and heat treated A1 and A2 alloys was studied and compared with base alloy A at an applied load of 20–60 N and sliding distances from 0 to 3000 m at a constant sliding speed of 200 RPM. Three tests were conducted on each material, and the average result was calculated. After each test, the wear disc and specimen were cleaned with acetone to ensure a clean surface without the wear debris. Wear (µm) was calculated and directly generated in the system. However, wear rates were calculated using mass loss method to validate the system-generated results. The wear data obtained from the system typically considers the effect of both the pin sample and the counterface, as both surfaces are in contact and can influence each other's wear behavior. Therefore graphs were plotted considering the wear results obtained from the system directly. Each

specimen's initial and final weight before and after the test were noted to calculate the weight loss (Δm). The wear rate was calculated using equation 1.

$$\text{Wr} = \frac{\Delta m}{dPS} \qquad (1)$$

where $W_r$ is the wear rate (mm$^3$/N.m), Δm is the weight loss (mg), P is the loading weight (N), S is the sliding distance (m), and d is the density (g/cm$^3$) [32].

### 3. Results and Discussion

*3.1. Microstructure Studies of As-Cast Alloys*

Microstructure analysis of as-cast A, A1, and A2 alloys was performed using a SEM microscope along with EDS analysis to quantify elements. Samples were polished for a mirror surface and etched for 25 s using Keller's reagent. Figure 4 shows the SEM microstructure images of the as-cast alloys.

The SEM microstructure of as-cast A, shown in Figure 4a, displays well-dispersed pro-eutectic aluminium and fine eutectic Al-Si colonies. The primary phase in the A specimen is α-Al (1), which forms a dendritic structure through precipitation of various eutectic reactions and a trace amount of alloying elements that dissolve in aluminium [33]. The grain size was large, with some pro-eutectic phases forming within the grains. As there is a lack of alloying elements to control grain size, the eutectic colony was observed to be coarse in this alloy.

The SEM images in Figure 4b reveal the microstructure of the as-cast A1 alloy. It consists of primary α-Al (1), a network of eutectic Si plates (2), and Mg$_2$Si (3). The presence of π-Al$_8$Mg$_3$FeSi$_6$ (4) and β-Al$_5$FeSi (5) intermetallic phases, commonly found in Al-Si alloys, is also noted. With 1 wt.% Mg added, all Mg atoms react primarily with Si to form Mg$_2$Si (3) and Al$_5$Cu$_2$Mg$_8$Si$_6$ (6) phases, while the remaining Mg atoms convert β phase into π phase [15,33,34]. The grain size is larger in A1 compared to A, with some pro-eutectic phase within the grain, and it may be due to the combined effect of Cu and Zn as trace elements.

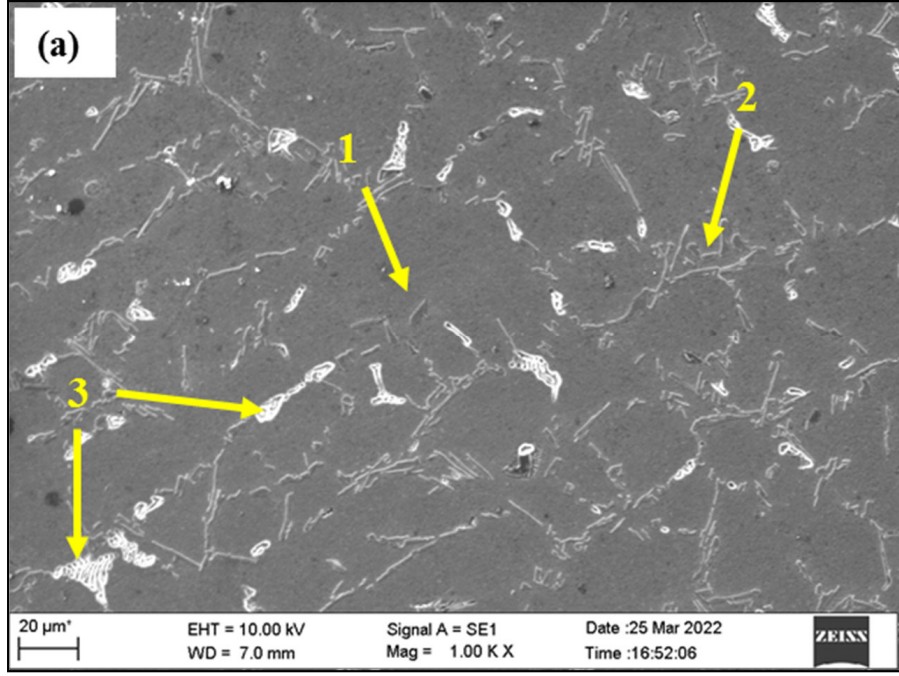

**Figure 4.** *Cont.*

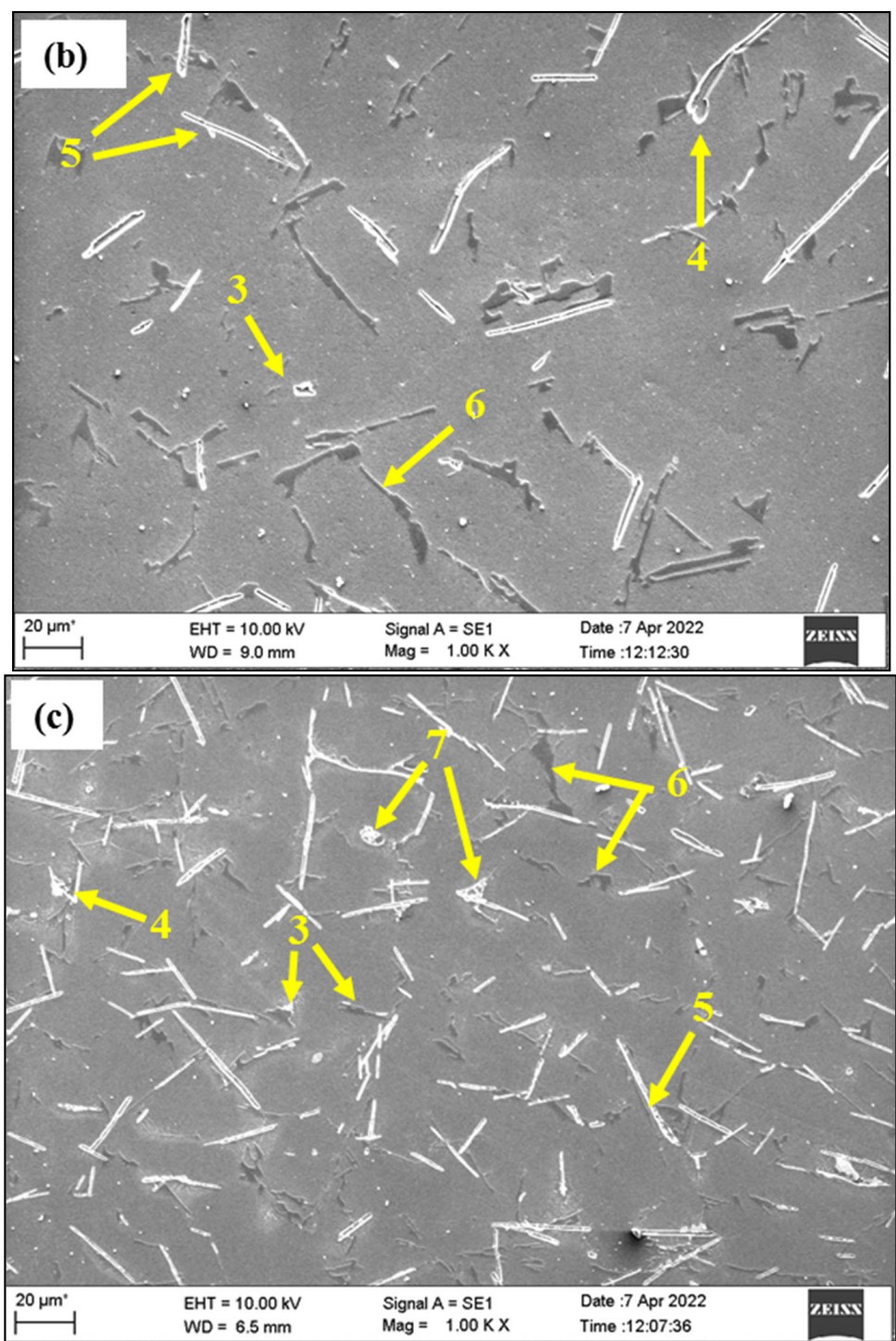

**Figure 4.** SEM images of as-cast alloys (**a**) alloy A, (**b**) alloy A1, and (**c**) alloy A2.

Figure 4c displays SEM images of the as-cast A2 alloy. The as-cast A2 microstructure includes α-Al (1), eutectic Si (2), Q- $Mg_2Si$ (3), Fe-rich phases π-$Al_8Mg_3FeSi_6$ (4), β- $Al_5FeSi$ (5), and S- $Al_5Cu_2Mg_8Si_6$ (6) phase [35,36]. Addition of 1 wt.% Cu to A2 alloy resulted in a hard $Al_2Cu$ (7) intermetallic phase, which is responsible for the increase in hardness value. Compared to A and A1, A2 alloy showed a greater amount of pro-eutectic α-Al, due to the high melting point of copper promoting precipitation hardening. The pro-eutectic phase is dispersed along and within the grain boundary, with an excellent dispersion compared to A and A1 alloys (Figure 4). No cavities or blow holes were observed, with a dominant directed solidification mechanism favouring dendritic formation.

The XRD analysis results of an as-cast A2 sample are displayed in Figure 5. The plot shows the presence of diffraction peaks of $\alpha$-Al and Si phases. The presence of $Al_2Cu$ and $Mg_2Si$ peaks is confirmed through the presence of 1 wt.% Cu in the A2 sample, supporting previous research findings [37,38].

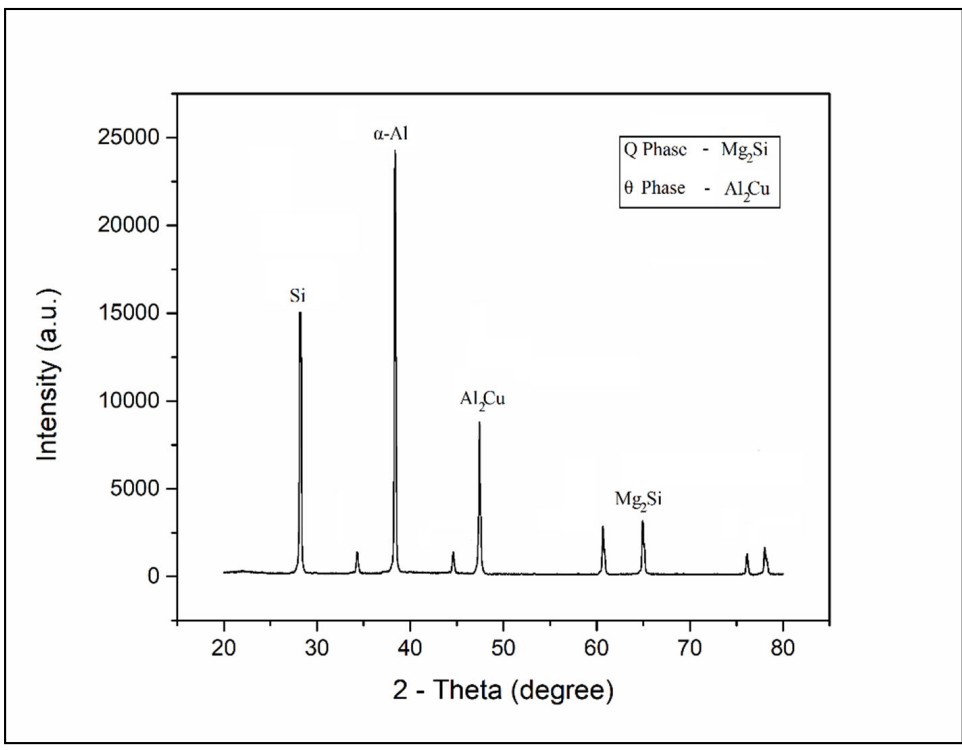

**Figure 5.** XRD analysis plot of A2 alloy.

### 3.2. Hardness of As-Cast A, A1, and A2 Alloys

To investigate the hardness variation on as-cast specimens A, A1, and A2 with minor additions of zinc and copper as alloying elements, micro Vickers hardness tests were performed. The results of all the hardness tests conducted on the as-cast alloy specimens A, A1, and A2 are recorded in the form of tables and graphs. The measured hardness values of the as-cast alloy samples are displayed in Table 4.

**Table 4.** Hardness values of as-cast A, A1, and A2 alloys.

| Designation | Vickers Hardness Number (VHN) | | | | | Average Vickers Hardness Number (VHN) |
|:---:|:---:|:---:|:---:|:---:|:---:|:---:|
| | Trail 1 | Trail 2 | Trail 3 | Trail 4 | Trail 5 | |
| **A** | 54.2 | 53.7 | 52.1 | 53.0 | 53.4 | **54** |
| **A1** | 61.7 | 62.3 | 61.1 | 62.5 | 61.9 | **62** |
| **A2** | 75.5 | 76.1 | 74.0 | 73.8 | 72.5 | **75** |

The increased hardness values in alloys A1 and A2 were due to modifications in the size and shape of eutectic silicon particles and the formation of intermetallics. The hardness of A, A1, and A2 alloys is 54, 62, and 75 VHN, respectively. Adding 0.5 wt.% Zn and 0.5 wt.% Cu to alloy A increased the hardness value by 16%, while 1 wt.% Cu addition increased it by 39% due to the formation of high-melting-point $Al_2Cu$ intermetallics. A2 alloy showed more significant hardness improvement than A1 due to a larger amount of copper addition. The small amount of zinc added to A1 may not have formed sufficient intermetallics to improve hardness.

The effect of copper addition to A356 on hardness improvement in the alloy was mainly due to the formation of hard $Mg_2Si$ and $Al_2Cu$ intermetallic phases as observed in the microstructure. The limited addition of 0.5 wt.% copper in A1 alloy did not result in a noticeable improvement in hardness because it does not form enough $Al_2Cu$ intermetallics phases. The increased hardness in A2 alloy with 1 wt.% Cu was attributed to the refinement of pro-eutectic α-Al dendrites, solid solution hardening, and the formation of $Al_2Cu$ phases.

### 3.3. Aging Hardening Treatment of A, A1, and A2 Alloys

All prepared samples A, A1, and A2 were subjected to an age hardening treatment (T6), and the peak hardness value was determined. Figure 6 depicts the hardness variation of T6-treated A, A1, and A2 alloys. The results clearly show that the hardness value increases with increasing time for all samples until the peak aged condition is reached. Holding the samples after peak aging reduced the hardness value, indicating an overaged condition. The time required to reach peak aged condition for alloys A1 and A2 decreases with an increasing weight percentage of copper. For samples aged at 100 °C, alloy A reached its peak hardness value of 64 VHN after 10 h, while the A1 and A2 samples showed peak hardness values of 95 and 107 VHN after 9 and 8 h, respectively. The significant increase in hardness of A1, which was 76% higher than A, was primarily attributed to the solid solution strengthening of Cu in the A356 matrix due to the formation of various intermetallics [39]. This showed that the addition of Cu primarily enhanced both the age hardening ability along with age hardening rate. The higher the weight percent of Cu, the greater the formation of intermetallics, leading to a 98% increase in hardness for A2 compared to A. Aging at 200 °C resulted in lower hardness values, probably due to aging kinetics [40].

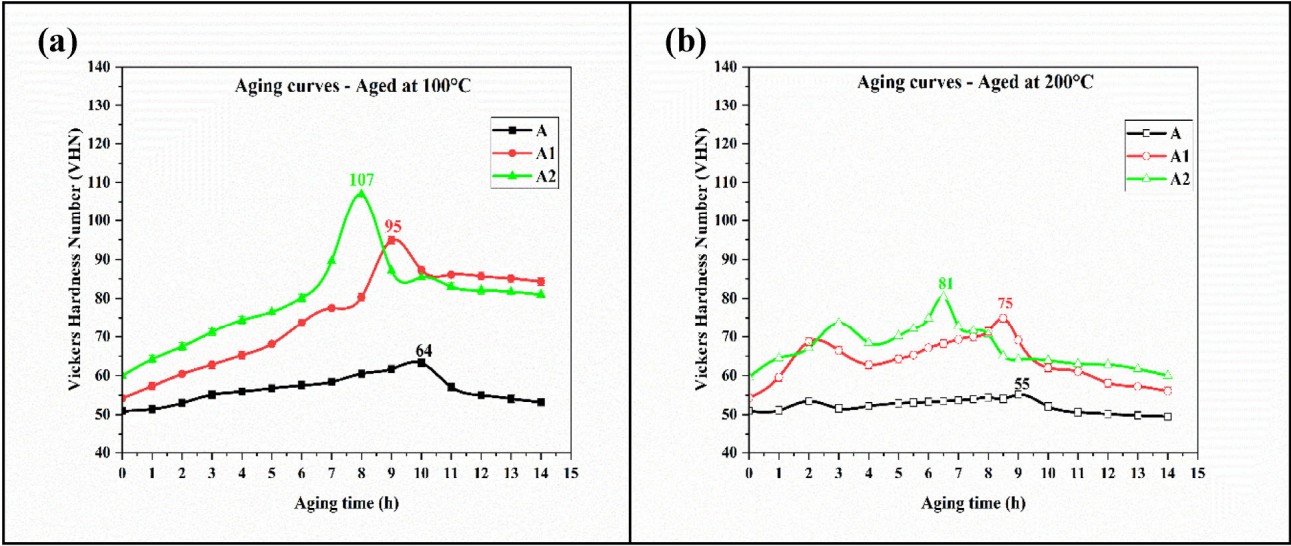

**Figure 6.** Aging curve of alloys (**a**) aged at 100 °C and (**b**) aged at 200 °C.

Adding Cu to A356 alloy improved its hardness due to solid solution strengthening during solutionising, with maximum hardness at peak aged conditions. Increasing Cu wt.% lead to an improvement in solid solution strengthening in A356. This was consistent with prior studies involving adding small amounts of copper to Al-Si based alloys. Additionally, 1 wt.% Mg added to A356 resulted in solid solution strengthening through $Mg_2Si$ intermetallic formation.

Figure 7 shows the peak hardness variation in A, A1, and A2 alloys. The time to reach peak aged condition in alloys A1 and A2, aged at 100 °C and 200 °C, decreased with increasing copper content. Zinc addition (0.5 wt.%) did not significantly impact hardness improvement at peak aged condition. Aging at 100 °C, as-cast A showed peak hardness of 64 VHN after 10 h, while A1 and A2 had peak hardness values of 95 and 107 VHN after

9 and 8 h, respectively. This improvement in hardness of A1 and A2 over A was due to $Al_2Cu$ intermetallic phase formation and increasing Cu content. Aging at 100 °C resulted in 76–98% improvement in hardness compared to as-cast A, while aging at 200 °C resulted in 39–50% improvement. Lower aging temperatures (100 °C) resulted in higher hardness values but a shorter time to reach peak hardness compared to higher temperatures (200 °C).

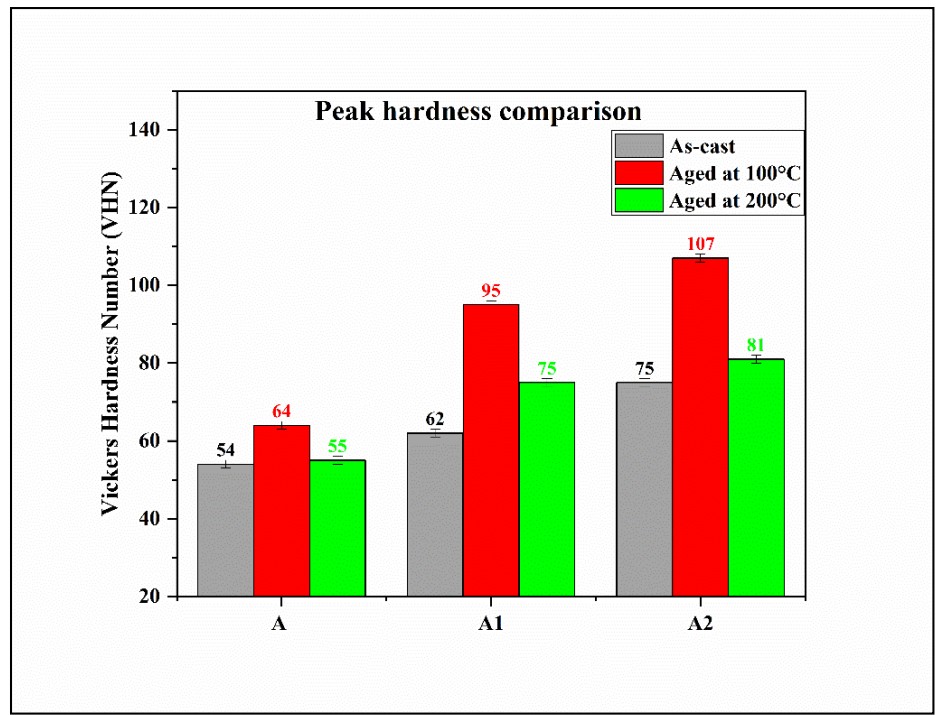

**Figure 7.** Peak hardness comparison of as-cast and T6 treated alloys.

### 3.4. Wear Studies of As-Cast and Age Hardened A, A1, and A2 Alloys

Alloys used for tribological applications must have high strength, low friction, and high wear resistance. Variables such as alloying elements, mechanical deformation, and heat treatment can improve specific strength, wear resistance, high-temperature strength, stiffness, and damping capacity [39,41–43]. Wear resistance is also affected by working conditions such as applied load, sliding distance, sliding speed, and temperature [44,45]. Adding alloying elements or reinforcements can enhance wear resistance in A356 alloy [46]. Figures 8–10 show the variation of wear with increasing sliding distance under different loads for alloys A, A1, and A2.

The results show that the wear rate of both as-cast and heat-treated alloys A, A1, and A2 increased with increased applied load, as per Archard's wear law [47]. In both as-cast and heat-treated conditions, the wear rate initially increased gradually up to 20 N, indicating mild wear characteristics, but it significantly rises when the applied load exceeds 40 N. At 60 N applied load, severe wear was observed with material transfer from the pin to the steel rotating disc. As-cast alloy A showed the highest wear of 476 μm at 60 N, indicating severe wear. In severe wear, a heavy metal transfer occurs from the pin to the counterface during sliding, resulting in higher mass loss and wear rate. In mild wear, the pin oxidizes and creates less friction, leading to a lower wear rate. On comparison of alloys A, A1, and A2, A2 alloy showed maximum wear resistance property at 100 °C peak aged condition.

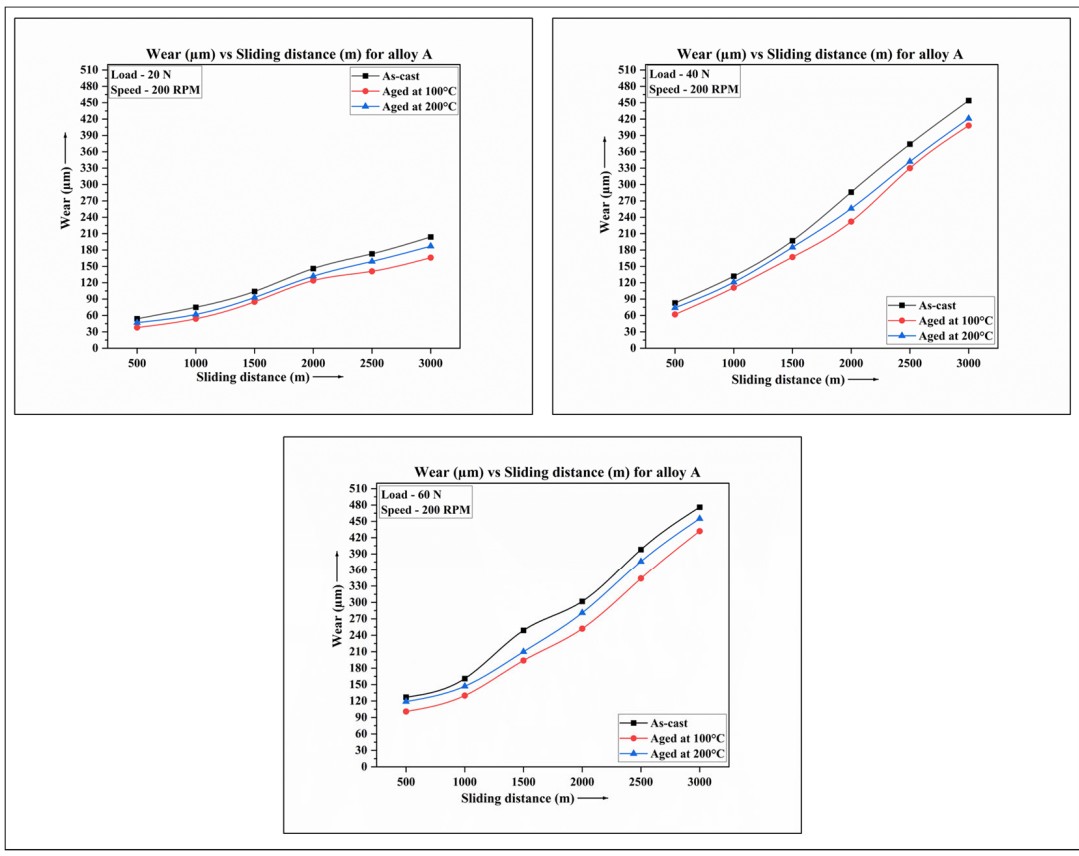

**Figure 8.** Variation of wear with sliding distance for as-cast and heat-treated alloy A under different loads.

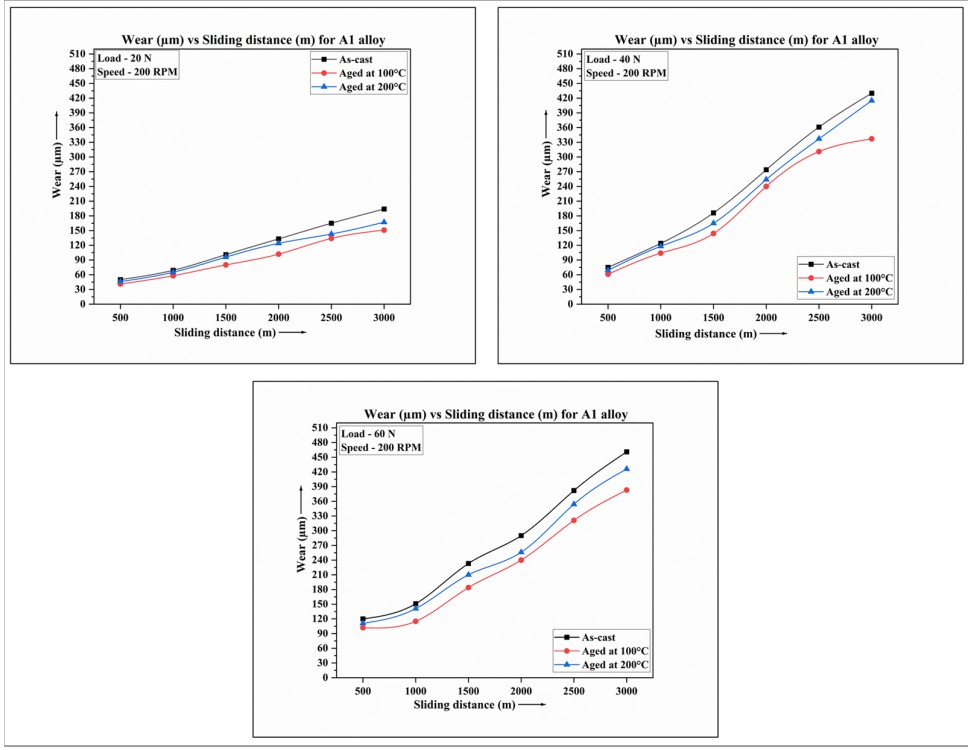

**Figure 9.** Variation of wear with sliding distance for as-cast and heat-treated alloy A1 under different loads.

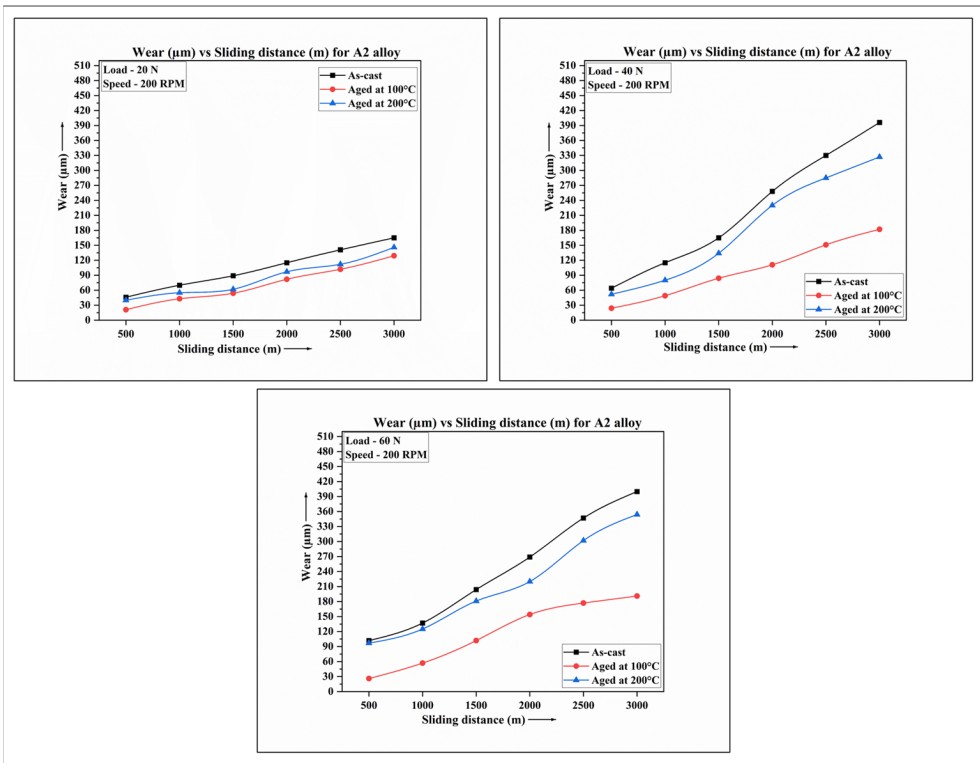

**Figure 10.** Variation of wear rate with sliding distance for as-cast and heat-treated alloy A2 under different loads.

The effect of sliding distance on wear rate of the alloys was also studied. As expected, the wear rate increases with increasing sliding distance in all the studied conditions. Under the studied as-cast condition of the alloys, A2 alloy showed the highest wear resistance at a higher sliding distance with wear ranging between 165 and 400 μm for 20–60 N applied load. This may be due to the effect of the addition of 1 wt.% Cu in A2 alloy where $Al_2Cu$ intermetallic phase was observed. However, A, A1, and A2 samples at peak aged conditions of 100 and 200 °C showed reduced wear rate when compared to as-cast samples. This may be attributed to the effect of grain refinement of α-Al and eutectic Si during the aging process and hardness improvement by the formation of hard $Al_2Cu$ and $Mg_2Si$ intermetallics. During sliding wear, the interactions between the dislocations and $Al_2Cu$ phase prevent crack propagation, which may have reduced the amount of material loss. However, under similar conditions of load and sliding distance, peak aged samples at 100 °C showed a much lower wear rate compared to as-cast samples and samples peak aged at 200 °C (aging kinetics). The 100 °C peak aged A2 alloy showed the highest wear resistance property, which is attributed to the addition of 1 wt.% Cu to alloy A. The addition of Cu resulted in the solid solution strengthening by precipitation hardening along with the formation of $Mg_2Si$ and $Al_2Cu$ intermetallics, thereby showing significant improvement in hardness and wear resistance properties in A2 alloy.

The friction coefficient characteristics of the wear surface are generally affected by factors such as load, temperature, and sliding distance [48]. The friction coefficient curve of A, A1, and A2 at different loads and sliding distances is shown in Figures 11–13.

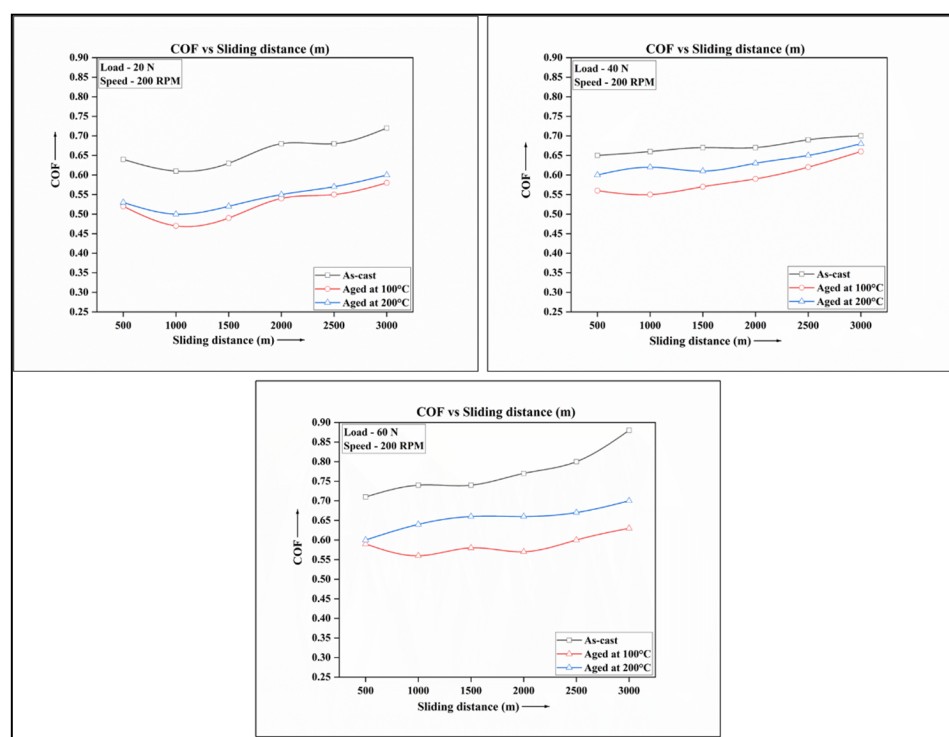

**Figure 11.** Variation of COF with sliding distance for as-cast and heat-treated alloy A under different loads.

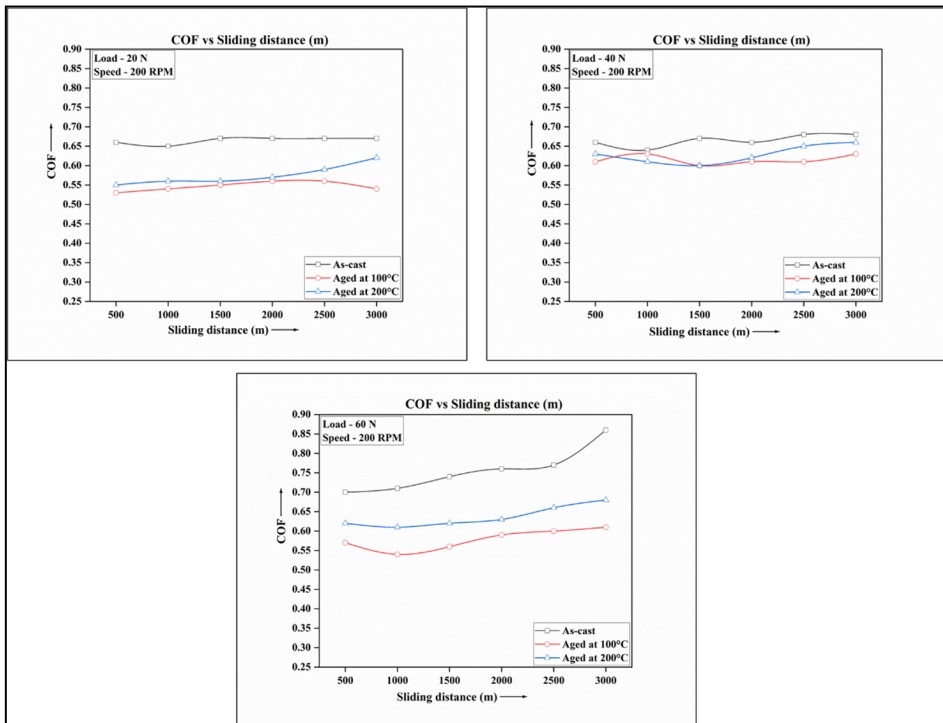

**Figure 12.** Variation of COF with sliding distance for as-cast and heat-treated alloy A1 under different loads.

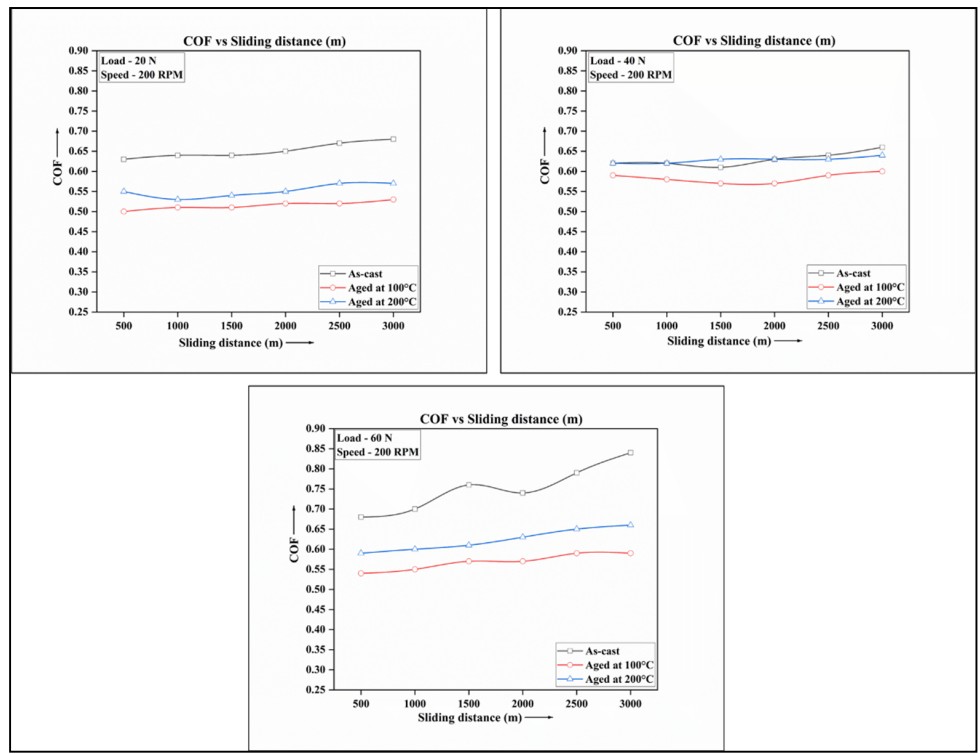

**Figure 13.** Variation of COF with sliding distance for as-cast and heat-treated alloy A2 under different loads.

Initially, the friction force is low at the start of the wear test due to the polished wear surfaces. However, once the wear surfaces become rough, friction force increases quickly. At a load condition of 20 N, the average coefficient of friction of as-cast A2 alloy was found to be 0.64 at a lower sliding distance of 1000 m. At the sliding distance of 3000 m, the average coefficient of friction reached 0.68 showing no significant increase in frictional force. This may be due to the formation oxide layer, which acts as a lubricant reducing frictional force and the amount of material loss during sliding action. The low shear strength and ductility of oxide layers cause them to remain intact at lower loads, reducing metal–metal contact, and leading to lower friction and wear rates. However, with increasing loads, the stability of these oxide layers is compromised, leading to metal–metal contact and damage, resulting in a rise in friction and wear rates [49]. At 60 N applied load, the friction coefficient started at 0.71 and started to fluctuate, reaching 0.82 at a sliding distance of 3000 m, as shown in Figure 13. This may be attributed to the hard abrasive debris which falls off from the worn surface during the test resulting in a large change in the surface roughness causing larger fluctuations. Over time, as the sliding distance increases, the frictional force increases significantly. With the increase in friction between the pin and the counterface, cracks appear on the surface, thus increasing the surface roughness and frictional force [50]. Under such conditions, as-cast alloy A1 showed a similar trend to as-cast alloy A with a lesser coefficient of friction. However, as-cast alloy A2 showed the lowest coefficient of friction, reaching a maximum coefficient of friction of 0.82 at an applied load of 60 N and sliding distance of 3000 m.

Compared to as-cast samples, peak aged samples exhibited much lower coefficients of friction, as shown in Figures 11–13. Peak aged 100 °C samples of A2 alloy exhibited the lowest coefficient of friction when compared to peak aged A and A1 alloys at 100 and 200 °C. The 100 °C peak aging of samples with the addition of 1 wt.% Cu as an alloying element to alloy A, i.e., A2 alloy, resulted in obtaining maximum wear resistance property.

The SEM images of the worn surfaces of as-cast A, A1, and A2 alloys were analysed and compared, as shown in the Figure 14. All alloys were tested for dry sliding wear under different applied loads (20–60 N) and at a constant sliding speed of 200 RPM and distance

of 3000 m, both in as-cast and heat-treated conditions. The SEM images showed that as-cast alloy A exhibited higher fracture compared to A1 and A2 alloys.

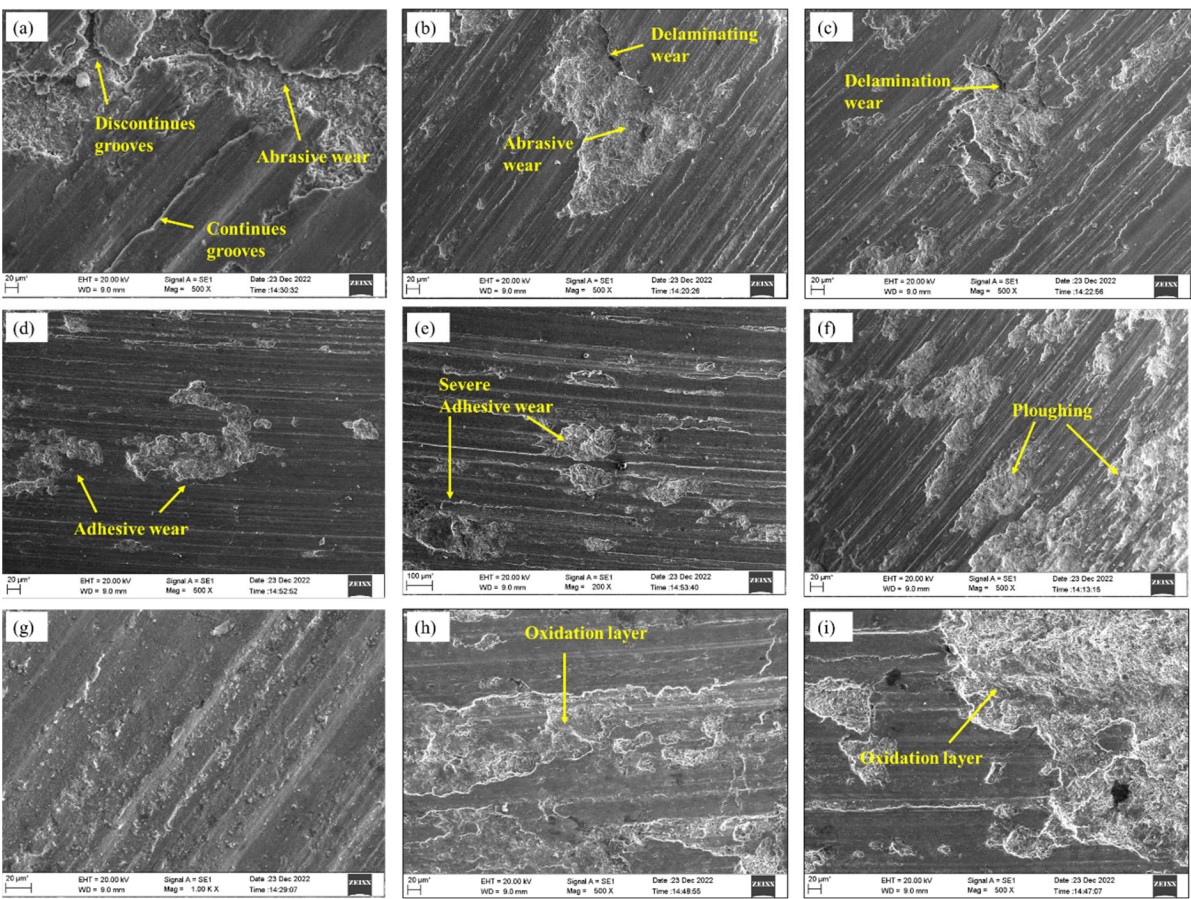

**Figure 14.** SEM images of worn surfaces. Worn surfaces of alloy A with different applied loads (**a–c**): (**a**) 20 N, (**b**) 40 N, and (**c**) 60 N. Worn surfaces of alloy A1 with different applied loads (**d–f**): (**d**) 20 N, (**e**) 40 N, and (**f**) 60 N. Worn surfaces of A2 alloy with different applied loads (**g–i**): (**g**) 20 N, (**h**) 40 N, and (**i**) 60 N.

Figure 14a shows the worn surface of as-cast alloy A under an applied load of 20 N. The SEM image of the worn surface reveals that it comprises both continuous and discontinuous grooves across the surface. This can be attributed to abrasive wear caused by particles trapped during sliding action. At this lower load, mild abrasive wear was the dominant wear mechanism. As the applied load increased to 40 N and 60 N, abrasive and delaminating wear became more prominent and caused more material loss and severe wear on the surface of the alloy, as shown in Figure 14b,c. Abrasive wear mechanism became the primary contributor to material loss and severe wear, with delaminating wear mechanism also contributing to surface layer separation and material loss. Figure 14d–f shows the worn surface of A1 alloy, which has furrows and spalling pits due to plastic deformation. At an applied load of 20 N (Figure 14d), wear is minimal, with a shallow surface and limited peeling, mainly caused by adhesive and mild fatigue wear. With a load increase to 40 N (Figure 14e), the wear mechanism changed to a mix of abrasive and severe adhesive wear, with bigger spalling pits and abrasive wear from debris. At 60 N (Figure 14f), wear intensifies with many pits, flakes, and internal cracks, primarily caused by delamination wear resulting from acting shear forces and friction heat [51,52]. Figure 14e,f shows the wear surface images of alloy A2 at an applied load of 20, 40, and 60 N, respectively. From the wear surface image, it is clear that the worn surface consisted of fine parallel abrasive groves along with an oxide layer running throughout the surface

indicating oxidative wear and mild abrasive wear mechanism as the predominant wear mechanism at lower load conditions. It is known that the oxide layer is formed by Fe-rich phases present in the counter disc material forming a thin covering layer that acts as a lubricant reducing the wear rate at higher load conditions (40–60 N). Energy dispersive X-ray (EDX) results confirmed the formation of an Fe-rich oxide layer on the A2 surface resulting in an improvement in wear resistance (Figure 15).

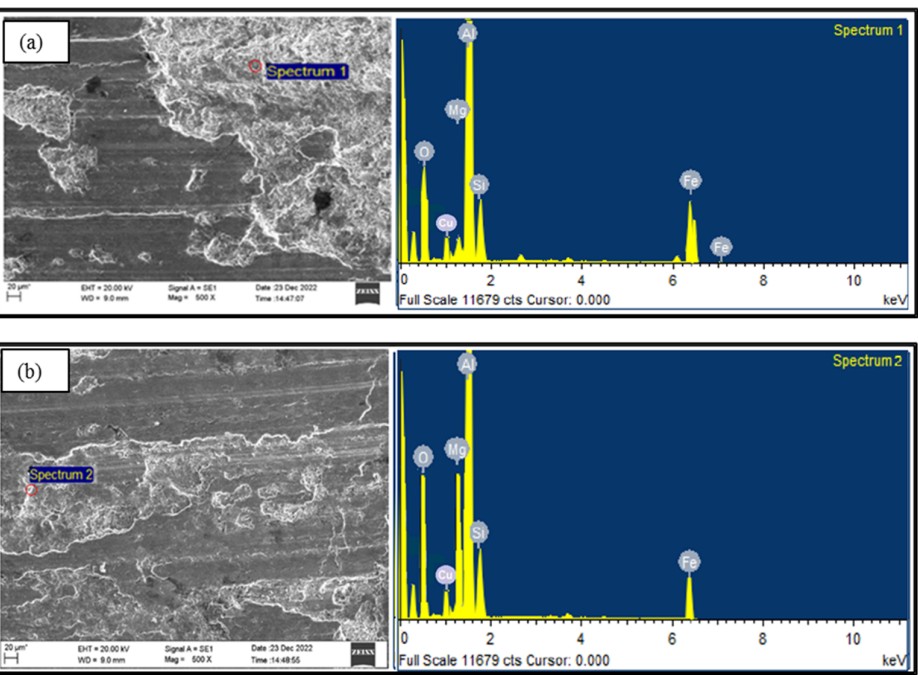

**Figure 15.** EDX results of A2 alloy at applied loads: (**a**) 60 N and (**b**) 40 N.

The transfer of material during the sliding process, along with surface delamination, results in the formation of wear debris. Wear debris appears dark in colour and was analysed under SEM. Figure 16 shows typical SEM micrographs of the wear debris from the extruded A, A1, and A2 alloys. Three types of morphologies were observed in the collected debris (large coarse plate-like flakes, medium coarse-like flakes, and fine particles) for A, A1, and A2 alloys. For alloy A, the debris comprised of larger and coarse-sized plate flakes with debris sizes ranging from 50to 200 μm being observed (Figure 16a). SEM images of A1 alloy showed a mixture of fine particles with shiny metallic plate-like flakes (Figure 16b). This shows us that for A1 alloy medium-sized wear debris was formed during the sliding process. However, alloy A2 showed very fine particles (Figure 16c). From the wear debris analysis, adhesion wear is confirmed to be the main wear mechanism based on the topography of the wear debris, which is in the form of flake-like sheets and irregular flakes at higher wear rates caused by material breaking off at the edge of the worn surface.

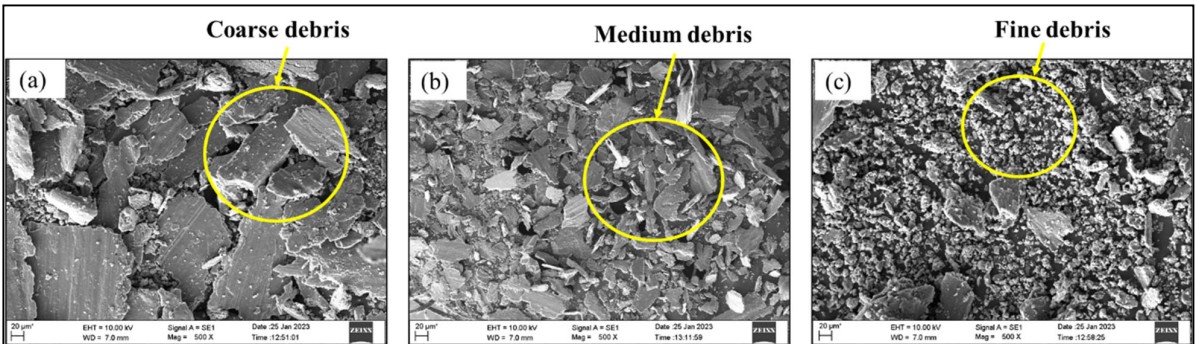

**Figure 16.** SEM images of wear debris at applied load of 60 N and 3000 m sliding distance: (**a**) alloy A, (**b**) alloy A1 and (**c**) alloy A2.

## 4. Conclusions

1.  The microstructure analysis revealed that under as-cast conditions the addition of zinc to the A356 alloy has no role in controlling grain size during the solidification process. On the other hand, the inclusion of 1% copper as an alloying element to A356 resulted in the formation of a greater amount of pro-eutectic α-Al, leading to finer eutectic colonies of Al-Si. Furthermore, the microstructure study found that adding only 0.5% copper to the A356 alloy may not be sufficient to produce a substantial amount of $Al_2Cu$ intermetallic phase. This information supports the conclusion that the presence of copper as an alloying element has an impact on the solidification process and the resulting grain structure, but a higher concentration of copper is necessary for the formation of a notable quantity of $Al_2Cu$ intermetallic phase.

2.  The highest hardness value of 75 VHN in as-cast alloy specimens was obtained with the addition of 1% copper (A2 alloy), and this was attributed to the precipitation of the $Mg_2Si$ and $Al_2Cu$ intermetallic phases. The addition of copper to the A356 alloy was found to enhance the age hardening properties when quenched and subjected to aging conditions through both solid solutions strengthening and the formation of various intermetallic phases. Aging of alloys at 100 °C displayed a 95% improvement in hardness at the peak aging stage, although the process took longer than aging at 200 °C.

3.  The 100 °C peak aging and 1 wt.% Cu as an alloying element to A356 alloy resulted in maximum wear resistance property. Improvement in wear resistance property in A1 (0.5 wt.% Zn + 0.5 wt.% Cu) and A2 (1 wt.% Cu) alloy compared to base alloy A356 can be summarized as follows:

    *   Precipitation hardening treatment and the addition of a minor quantity of zinc and copper as an alloying element to A356 alloy has resulted in the overall improvement in wear resistance with a lesser coefficient of friction.
    *   At lower load conditions (20 N), A1 (0.5 wt.% Zn + 0.5 wt.% Cu) alloy showed 15–22% increase in wear resistance aged at 200 °C and 29–35% increase in wear resistance aged at 100 °C, whereas A2 (1 wt.% Cu) alloy showed 36–40% increase in wear resistance aged at 200 °C and 58–75% improvement in wear resistance property aged at 100 °C.
    *   At higher load conditions (40–60 N), A1 (0.5 wt.% Zn + 0.5 wt.% Cu) alloy showed 6–14% increase in wear resistance aged at 200 °C and 30–40% increase in wear resistance aged at 100 °C, whereas A2 alloy showed 29–35% increase in wear resistance aged at 200 °C and 150–182% improvement in wear resistance property aged at 100 °C.

4.  Worn surface analysis of A356 + 1 wt.% Cu (A2) alloy confirmed the presence of an oxidation layer between the mating surfaces, which resulted in a reduction in wear rate and coefficient of friction both at lower and higher load conditions.



**Author Contributions:** Conceptualization, S.S. and G.M.C.; methodology, G.M.C., K.B.M., R.N., M.S. and N.K.; investigation, M.S., N.K. and G.M.C.; data curation, N.K., S.S., G.M.C. and R.N.; writing—original draft preparation, N.K. and S.S.; writing—review and editing, S.S., R.N., K.B.M. and G.M.C.; supervision, S.S. project administration, S.S., M.S. and G.M.C. All authors have read and agreed to the published version of the manuscript.

**Funding:** This research received no external funding.

**Data Availability Statement:** Not applicable.

**Acknowledgments:** The authors would like to acknowledge Manipal Academy of Higher Education, Manipal for providing the infrastructural facility to conduct the experimental works.

**Conflicts of Interest:** The authors declare no conflict of interest.

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
