# Peer review of "Experimental Investigation of Mechanical Property and Wear Behaviour of T6 Treated A356 Alloy with Minor Addition of Copper and Zinc"

_jcs, doi:10.3390/jcs7040149_

Round 1

Reviewer 1 Report

K. Nithesh and his colleagues present a study on the effect of including small quantities of Cu and Zn on strengthening the A356 (Al-7Si) alloy using the so-called T6 treatment.

The paper presents the experimental work in a concise and clear manner. The techniques used (Pin-On-Disc tribometer, SEM, XRD, Vickers hardness test) combine to yield relationships between important parameters and the microscopic configuration of the phase structure. All the information is presented in tables, providing a set of data for each experiment.

Overall, this paper presents interesting information for the community working on the tribological properties of aluminum alloys, and I recommend its publication in the Journal of Composites Science.

Author Response

Dear Reviewer 1,

Thank you for your positive comments and acceptance of the article.

Reviewer 2 Report

Although the article presents fairly up-to-date engineering issues, the paper in its current state is not suitable for publication. Rather, the work is academic in nature and presentation. In my opinion, the way the results are presented determines the unpublishability of the research.

Author Response

The reviewer has not suggested any specific queries or questions to be answered. Hence, we have answered the queries of the other reviewers.

Thank you.

Reviewer 3 Report

Experimental study on mechanical properties and wear resistance of T6 treated A356 alloy with a small amount of copper and zinc

1 Research content and significance

Typical applications of A356 alloy include aircraft engine and pump parts, fuselage and landing wheel, truck chassis parts, aircraft parts and control parts, and structural elements requiring high strength. Although it has excellent properties, the addition of various alloying elements, such as magnesium (Mg), copper (Cu) and zinc (Zn), can effectively improve the properties of the alloy, and will also affect the wear behavior of A356 alloy due to precipitation hardening and solid solution strengthening. In addition, these small amount of alloy elements such as Mg, Cu and Zn enter the Al solid solution and form various strengthening phases under different conditions.

Referring to a large number of previous studies, it is found that the research work on improving the wear performance of A356 alloy by adding a small amount of copper, zinc and magnesium is limited. Therefore, in this article, the author tries to add a small amount of copper, zinc and magnesium as alloy elements into A356 to determine the changes of microstructure, hardness and wear behavior under as-cast and aging hardening conditions. Through research, the author found that the formation of Al2Cu intermetallic phase requires at least 1 wt.% Copper, resulting in finer grain structure and improved hardness. During peak aging at 100 ° C, the highest hardness of 107 VHN (increased by 98%) can be obtained by adding 1 wt.% copper. In addition, the wear test showed that 1 wt.% was added The copper and precipitation hardening (T6) treatment at 100 ° C of A356 alloy increased the wear resistance by 150-182% and reduced the friction coefficient. The wear surface analysis shows that there is an iron-rich oxide layer at the contact interface of the sample, which improves the tribological properties of A356 alloy. These conclusions have positive guiding significance for the research and development of lightweight materials that can be used to manufacture various automotive and aerospace components, and guide the exploration of high-performance materials.

2 Suggestions on article content

A In this paper, three A356 alloys with different compositions in the following table are used as experimental samples to compare and study the effect of added elements on their properties.

I think the existence of 0.5wt.% Zn in A1 and A2 samples makes the conclusion unconvincing and cannot prove that the improvement of alloy properties is caused by the difference of Zn or Cu content. Add a group of samples A3: 1 wt.% Mg+0.5 wt.% Zn+0.5 wt.% Cu on the original basis, and the conclusion may be clearer and more intuitive.

B When inserting Figure 7, adjust the alignment of the text and the picture to make it more beautiful.

C The insertion mode of Figure 9 is changed to be the same as that of Figure 10 and Figure 11. The information in the figure can be expressed more clearly.

3 Comprehensive evaluation

This article uses (T6) to treat alloys with different added elements. The research purpose is clear and the method is innovative. Compared with other methods, the method used in this article is simple and convenient. Its test conditions show that this method has potential value in practical application. The experimental conditions have been carefully optimized and the alloy has been tested under various practical conditions. The content is substantial, but the contrast of the experimental sample composition is slightly weak. It is suggested to add a group of samples to make the conclusions more clear and convincing. The whole article has clear ideas, and the data listed can well support the corresponding conclusions, which is of guiding significance for the exploration of high-performance alloys. It is recommended to agree to accept after modification.

Reviewer 4 Report

The present paper has studied on effects of copper and zinc addition on hardness and wear/friction properties of T6 treated A356 alloy. The authors found that hardness and wear/friction properties could be improved by addition of cupper with an amount higher than 1wt%. The research topics is within the scope of Journal of Composites Science. However, there are several points to be clarified and revised according to the reviewer’s comments indicated below before considering whether the present paper can be accepted for publication.

(1) In the section 2.1, three kinds of materials have been prepared: A with 1wt%Mg, A1 with 1wt%Mg+0.5wt%Zn+0.5wt%Cu, and A2 with 1wt%Mg+1wt%Cu. If the effect of Cu addition is investigated, 0.5wt%Zn addition would be needless, which makes the discussion simple since the effect of Zn addition can be neglected. Please explain in the revised manuscript what is the purpose of 0.5wt%Zn addition to A1.

(2) In the section 2.4, the authors used two kinds of wear rate: one is the wear rate (um) generated by the wear test system and the other is the wear rate (mm3/N.m) defined by the equation (1). However, in the section 3.5, the wear rate (um) was adopted in the figures. The authors should confirm the validity and equivalence of the wear rate (um) to the wear rate (mm3/N.m) by using the specimen surface area (mm2) and the loading weight (N) and the sliding distance (m) as the wear rate = the wear rate (um) x specimen surface area (mm2) / the loading weight (N)/the sliding distance (m). If the wear rate (um) generated by the wear test system is not reliable, the wear rate (mm3/N.m) should be adopted in the figures instead of the wear rate (um).

(3) In the section 3.1, three kinds of microstructures are shown in Fig. 4. Based on these observations, average grain size, volume fraction and mean spacing of precipitated intermetallic particles should be listed in a table. They are dominant microstructural factors to influence wear and friction behavior as well as hardness.

(4) In the section 3.3, the results of hardness measurements are shown both in Table 4 and Fig. 6, which are duplicated. Fig. 6 should be deleted.

(5) In the section 3.4, the authors claimed that formation of intermetallic phase and grain refinement were induced by aging. It is better to show the evidence, which is the microstructure after aging. The average grain size, volume fraction and mean spacing of precipitated intermetallic particles after aging are also indicated in a table, similar to the comment (3).

(6) In the section 3.5, Figs. 9-11 and Tables 5-7 are duplicated. The tables should be deleted.

(7) In the last sentence of the paragraph below Fig. 14, the authors explain, “However, as-cast alloy A2 showed lowest coefficient of friction of 0.43 at the applied load of 60N and the sliding distance of 3000m”. However, the value of 0.43 is not consistent with Fig. 14.

(8) About the worn surface observations shown in Fig. 15, the surface damages on worn surfaces in A2 alloy looks the most severe at any loading weights compared to A and A1 alloys, which is not consistent with the authors’ claim about wear behavior of three materials. The related discussion should be modified in the revised manuscript.

(9) About the morphology of wear debris shown in Fig. 17, the authors discussed the reason of different morphology of wear debris only based on wear mechanism. However, from the purpose of the present study, the authors should discuss it also from the viewpoint of average grain size, volume fraction and mean spacing of intermetallic particles, which are induced by addition of Cu and Zn and already pointed out in the comment (3).

(10) About the conclusion 1, the authors claimed Zn has no influence on grain size. However, in the third paragraph of the section 3.1, the authors claimed that the grain size is large in A1 compared to A and the low melting point of Zn contributes to the coarser eutectic colony. Please make them consistent. 

(11) In the conclusion, based on the purpose of the present study, it should be clearly presented that what kinds of change of microstructure are induced by adding Zn and Cu elements and then how these microstructural changes influence hardness, wear rate and friction coefficient of the materials.

(12) About Abstract, the last sentence is not important. It is better to rearrange the whole abstract based on the viewpoint of the comment (11).

(13) English grammar is not always correct and should be corrected.

Round 2

Reviewer 2 Report

The article presents a style of rather academic work but does not necessarily present scientific research in an insightful way. In my opinion, it is not suitable for publication.

Reviewer 3 Report

it can be  accepted

Reviewer 4 Report

It is better to explain in the revised manuscript why the wear (um) is adopted in Figs. 8, 9 and 10 instead of the wear rate.